# Adherence to the Mediterranean Diet Improves Fatty Acids Profile in Pediatric Patients with Idiopathic Nephrotic Syndrome

**DOI:** 10.3390/nu13114110

**Published:** 2021-11-16

**Authors:** Turolo Stefano, Edefonti Alberto, Morello William, Bolzan Giulia, Syren M. Louise, Tamburello Chiara, Agostoni Carlo, Montini Giovanni

**Affiliations:** 1Pediatric Nephrology Dialysis and Transplant Unit, Fondazione IRCCS Ca’ Granda-Ospedale Maggiore Policlinico, Via della Commenda 9, 20122 Milan, Italy; alberto.edefonti@policlinico.mi.it (E.A.); william.morello@policlinico.mi.it (M.W.); giuliabolzan92@gmail.com (B.G.); chiara.tamburello@oliclinico.mi.it (T.C.); giovanni.montini@policlinico.mi.it (M.G.); 2Department of Clinical Sciences and Community Health, University of Milan, 20122 Milan, Italy; eva.syren@unimi.it (S.M.L.); carlo.agostoni@unimi.it (A.C.); 3Pediatric Intermediate Care Unit, Fondazione IRCCS Ca’ Granda Ospedale Maggiore Policlinico, 20122 Milan, Italy

**Keywords:** Mediterranean diet, nephrotic syndrome

## Abstract

The fatty acid profiles of patients with idiopathic nephrotic syndrome (INS) are different from that of healthy controls, even during remission, revealing an increase of the pro-inflammatory omega 6 series. It is still unknown whether the concomitance of nephrotic syndrome affects the potential positive effects of the Mediterranean diet on the levels of omega 3 and 6 fatty acids. We performed a cross-sectional study to evaluate the association between the adherence to the Mediterranean diet and fatty acid profile in 54 children with INS. The dietary habits were assessed through the validated Kidmed questionnaire. Patients with higher adherence had lower levels of linoleic acid and total omega-6. Moreover, a negative correlation between proteinuria and the anti-inflammatory omega-3 series was found. In conclusion, patients with INS with proteinuria and low adherence to the Mediterranean diet have an imbalance in the omega-6/omega-3 ratio that may benefit from following the Mediterranean diet.

## 1. Introduction

Idiopathic nephrotic syndrome (INS) is the most common glomerular disease in children, characterized by the triad of oedema, proteinuria and hypoalbuminemia. The underlying biological mechanisms are not fully understood, but in most cases, they probably involve immunological processes [1].

Fatty acid profiles of subjects with INS, even during remission, is different from that of healthy controls [2], particularly with regard to omega-6 fatty acids levels, whose increase may indicate a state of persistent latent inflammation, ultimately dependent on the well-known imbalance of lymphocyte sub-populations [3]. A correct dietary balance between omega-6 and omega-3, in favor of the latter, is essential both in healthy subjects and in those with kidney disease [4].

The Mediterranean diet is one of the three more healthy diets in the world, as adherence to it positively correlates with the levels of omega 3 eicosapentaenoic acid (EPA) and docosahexaenoic acid (DHA), which are known for their anti-inflammatory properties [5]. Moreover, in the general population, the Mediterranean diet is also able to counteract hypercholesterolemia, which is, however, a biochemical marker of nephrotic syndrome.

KidMed [6] is a questionnaire designed to assess adherence to the Mediterranean diet in pediatric subjects. It is a survey composed of 16 simple questions, and has been extensively validated by the scientific literature [7].

It is still largely unknown whether the concomitance of INS in each subject affects the potential positive effects of the Mediterranean diet on the levels of omega 3 and 6 fatty acids.

The aim of this study is to evaluate if adherence to the Mediterranean diet can influence the fatty acids profile in pediatric patients with INS, during proteinuria and remission, and if there is a correlation among patients’ fatty acid pattern and the typical biochemical parameters of INS.

## 2. Patients and Methods

### 2.1. Patients

We performed a single-center, cross-sectional study, which included all pediatric (<18 years old) patients with INS who underwent a routine blood test at the Pediatric Nephrology, Dialysis and Renal Transplant Unit of Fondazione Cà Granda IRCCS Ospedale Maggiore Policlinico, Milan, Italy, between 27 October 2020 and 8 January 2021. The inclusion criteria were pediatric patients (<18 years old) with diagnosis of INS, as specified below. Patients whose parents/legal guardians did not consent to the study were excluded.

Patients were enrolled according to the Helsinki declaration statement.

### 2.2. Definition of INS

Idiopathic nephrotic syndrome was defined by the presence of oedema, proteinuria >40 mg/m^2^/h in a 24  h urine collection, or a spot urine protein to creatinine ratio (uPr/uCr) of >2 mg/mg, and albuminemia <2.5 g/dl, in the absence of secondary causes of NS. Patients were further classified as steroid resistant or steroid dependent according to international guidelines [8].

### 2.3. Adherence to Mediterranean Diet

Adherence to the Mediterranean diet was evaluated by means of the Kid Med questionnaire. Patient’s parents or legal guardians were asked to fill the questionnaire in during the visit. The questionnaire consists of 16 questions on eating habits, each to be answered Yes (score: 1 or−1) or No (Score: 0). A total score is eventually calculated, whose maximal and minimal values range from −4 to12.

### 2.4. Fatty Acid Analysis

A blood sample of 200 μL was collected during routine checks. An aliquot of 50 μL was transferred into vials, methylated with 800 μL of HClMe 3N (Sigma Aldrich, Schnelldorf, Germany), incubated for 1 h at 90 °C and then refrigerated ad 4 °C for 10 min. Afterwards, 2 mL of KCl solution (Sigma Aldrich, Schnelldorf, Germany) and 330 μL hexane (Sigma-Aldrich, Schnelldorf, Germany) were added. Samples were first vortexed and then centrifuged at 3000 rpm for 10 min. Finally, the hexane layer (the upper layer) was collected from each vial and transferred into a gas chromatography vial for fatty acids profile evaluation with a Shimadzu Nexis GC-2030 (Shimadzu, Japan) gas chromatographer equipped with a 30 m fused silica capillary column FAMEWAX Restek (Restek, Centre Country, PA, USA). The gas chromatography results were analyzed using Labsolution software (Shimadzu, Japan).

Single fatty acids were expressed as relative percentage of total fatty acids.

Total saturated fatty acids (SFA), total monounsaturated fatty acids (MUFA), total polyunsaturated fatty acids (PUFA), total omega-3 (N3) and total omega-6 (N6) were also calculated. 

### 2.5. Biochemical Analyses

Urinary protein (UPr), urinary creatinine (UCr), serum triglycerides, total cholesterol and HDL cholesterol were measured and the uPr/Cr ratio was calculated as part of patients’ routine check analyses.

### 2.6. Statistical Analysis

Data were correlated by two tailed Pearson bivariate analysis; PCA analysis was performed to confirm results. The paired T-test was used to compare fatty acid profiles between different patient groups. *p*-values < 0.05 were considered statistically significant. Statistical analysis was performed with software SPSS 21 (IBM).

### 2.7. Ethical Approval

The study protocol was approved by a local Ethical committee with document number 0035199-U.

## 3. Results

### 3.1. Patients

A total of 54 patients (mean age 11 years ± 4; 27 males, 27 females; mean body weight 23.1 ± 8.6 kg) were enrolled: 44 of them were of Italian origin, three of Central/South American origin, six of Northern African origin and one of Asian origin. 37 patients were steroid dependent, 29 were in remission while eight were in relapse and were receiving treatment with steroid-sparing drugs including mycophenolate mofetil and calcineurin inhibitors. 1Seventeen patients were initially steroid resistant: nine of them subsequently responded to calcineurin inhibitors and were in remission, while eight who did not respond to immunosuppressant medications were classified as multidrug resistant and were proteinuric.

Table 1 summarizes the biochemical data of the patients according to the presence or absence of proteinuria. As expected, patients with proteinuria had significantly higher levels of total cholesterol and lower levels of serum protein and albumin than those in remission.

### 3.2. Adherence to Mediterranean Diet

48 out of 54 patients answered the questionnaire.

Patients’ adherence to the Mediterranean diet was in the medium-high quartile, with an average score of 5.13 ± 2.2 (minimum value 1, maximum value 9, on a scale from −4 to 12). In detail, the number of patients who responded Yes or No to each Kidmed question is summarized in Table 2. 

The population was then divided into two groups according to the KidMed score value (taking into account that the mean KidMed score was 5.1): those with high adherence (KidMed score ≥ 6), and those with low adherence (KidMed score < 4). All the cases with a score of 5) were excluded from the analysis.

No significant differences in Kidmed scores were found across different clinical groups: steroid dependent nephrotic syndrome (SDNS) vs. steroid resistant nephrotic syndrome (SRNS); Proteinuria vs. Remission; SDNS in remission vs. SDNS with proteinuria; SRNS in remission vs. SRNS with proteinuria) (Table 3).

Additionally, we didn’t detect any significant difference (*p*-value 0.53). in Kidmed scores between males (score 5.3 ± 2.1) and females (score 4.9 ± 2.3) of our cohort.

### 3.3. Fatty Acids Profile

When the population was divided into two groups according to the grade of adherence to the Mediterranean diet, the fatty acid profiles were significantly different with regard to linoleic acid and omega 6 levels, which were higher in subjects with low adherence, and saturated 22: 0 and 24: 0, which were higher in patients with high adherence. Total omega-3 and DHA levels were not statistically different in the two groups, but there was a tendency for both to be higher in patients with high adherence (Table 4).

The KidMed score values also showed a negative correlation with the levels of omega-6 (R^2^ −0.32; *p*-value 0.02) and a positive correlation with the saturated fatty acids (SFA) 22:0 e 24:0 (R^2^ and *p*-value 0.28/0.04 and 0.31/0.03 respectively). 

### 3.4. Correlations of Biochemical Parameters with KidMed Score and Fatty Acids

No significant correlations were found between KidMed score values and all biochemical parameters of INS.

As regards fatty acids, proteinuria, expressed as uPr/uCr, negatively correlated with the omega 3 DHA (R^2^ −0.32, *p*-value 0.03), with the total omega-3 (R^2^ −0.35, *p*-value 0.02) and with the saturated fatty acid18:0 R^2^ −0.3, *p*-value 0.04). This data was confirmed by PCA analysis (See Appendix A).

There was a negative correlation between triglycerides and arachidonic acid (R^2^ −0.39 *p*-value 0.012), the SFA 24:0 (R^2^ −0.35, *p*-value 0.02) and total PUFAs (R^2^ −0.32 *p*-value 0.04). Inversely, a positive correlation was found between triglycerides and the MUFA oleic acid (R^2^ 0.42, *p*-value 0.006) and total MUFAs (R^2^ 0.46, *p*-value 0.002).

Total cholesterol negatively correlated with arachidonic acid (R^2^ −0.36 *p*-value 0.02) and 18:0 (R^2^ −0.32 *p*-value 0.04), while it positively correlated with linoleic acid levels (R^2^ 0.36 *p*-value 0.019).

Total serum protein showed a positive correlation with 18:0 (R^2^ 0.44 *p*-value 0.03) and SFA (R^2^ 0.3 *p*-value 0.04) while serum albumin showed a positive correlation with 18:0 (R^2^ 0.42 *p*-value 0.04) and arachidonic acid (R^2^ 0.35 *p*-value 0.01)

## 4. Discussion

To the best of our knowledge, there is no data in the literature on the relation of the Mediterranean diet to nephrotic syndrome, despite the potential positive effects of the Mediterranean diet on various biochemical parameters (which are notoriously changed during INS), such as fatty acids, cholesterol and triglycerides.

Therefore, this study is the first which aimed to evaluate the dietary aspect of adherence to the Mediterranean diet in an important and widespread childhood renal disease like INS.

Our data show that this pediatric cohort, predominantly composed of Caucasian subjects of Italian origin, had a medium-high adherence to the Mediterranean diet, even if a small percentage of subjects (10%) significantly deviated, regardless of their ethnicity. The consumption of industrial snacks at breakfast (29 out of 48 subjects) and skipping breakfast (12 out of 48) impacted most negatively on the Kidmed score. 

As expected, adherence to the Mediterranean diet, as evaluated by the Kidmed score, was independent of patients’ clinical classification of INS and the presence of proteinuria. An analysis by sex and gender was conducted, even if previous studies did not find sex and gender differences in the adherence to the Mediterranean diet [7]. No significant difference was found in our cohort as regards the score on adherence to the Mediterranean diet for males compared to females.

In this study, adherence to the Mediterranean diet mainly influenced the blood levels of linoleic acid and total pro-inflammatory omega-6s, which were higher in patients with lower adherence than in those with higher adherence. 

This is consistent with the correlation reported in the literature between maternal breast milk composition in omega 6 and adherence to the Mediterranean diet [9].

On the contrary, levels of omega 3s were higher, even if not significantly, in patients with higher Kidmed score than in those with a lower score. This data was confirmed in the group of patients without proteinuria, where the 11 patients who had a Kidmed score of more than 6 showed a higher level of omega-3 and DHA than the 8 who had a Kidmed score less than 5 (eleven vs. eight). We could not confirm this data in the population with proteinuria (nine and four patients, respectively), 

Moreover, in our population, the levels of DHA and total omega-3, which are known for their anti-inflammatory properties, negatively correlated with proteinuria. As a consequence, we may hypothesize that a low level of these PUFAs contributes to INS progression and delay the remission of proteinuria.

Therefore, poor adherence to the Mediterranean diet and the state of proteinuria resulted in an increase of omega-6 fatty acids, in particular18:2n6 (linoleic acid), and a decrease of omega-3 and DHA, resulting in an imbalance of the N6/N3 ratio, in favor of a pro-inflammatory state.

As regards the increase of blood linoleic acid in patients with low adherence to the Mediterranean diet and the issue of possible dietary measures, it should be pointed out that linoleic acid is not considered at present a negative micronutrient per se. In effect, it has been established that its moderate intake is able to reduce total cholesterol and LDL concentrations [10] and to prevent the risk of development of cardiovascular disease [10,11]. However, the role of omega-6 metabolites is clearer, because they are involved in the inflammatory process in response to air pollution [12], and cause a shift in the microbiota, contributing to an increase of colonic inflammation [13]. Finally, a higher omega-6 dietary intake has been associated to an increase of pro inflammatory metabolites [14].

The importance of respecting a correct n-6/n-3 ratio in the diet, for the purpose of preventing chronic kidney disease, is essentially linked to the anti-inflammatory functions of linolenic acid (ALA, 18:3n-3) and longer-chain n-3 PUFAs [14]. According to this point of view, it would be advisable to reduce the amount of omega-6 in the diet, particularly in patients with nephrotic syndrome, in order to improve the dietary n6/n3 ratio, As a matter of fact, it has been recently demonstrated that patients with INS showed an abnormal fatty acid profile and high also arachidonic acid blood levels during remission, which allows for the hypothesis of a persistent inflammatory state in this population, even in the absence of proteinuria.

High cholesterol and triglycerides levels were inversely correlated with arachidonic acid, and positively correlated with MUFAs. The inverse correlation with arachidonic acid could be due to the fact that hypercholesterolemia causes an increase in the production of LTB4 [15], a metabolite of arachidonic acid, with the consequent decrease of arachidonic acid levels. As regards MUFAs, several studies in animal models have shown that high MUFA levels are associated with an increase in total serum cholesterol [16,17].

Interestingly, the absence of correlation between adherence to the Mediterranean diet and total cholesterol suggests that the Mediterranean diet is not able to effectively counteract the hypercholesterolemia of proteinuric patients with INS, as is the case of healthy subjects [18]. This could be due to the fact that the hepatic compensation mechanism of hypoalbuminemia, which is responsible for hypercholesterolemia, is persistent over time and dissociated from diet.

## 5. Conclusions

In patients with INS, better adherence to the Mediterranean diet modifies the omega-6/omega-3 ratio in favor of the anti-inflammatory omega-3s and reduces blood linoleic acid levels. This data suggests that patients with INS, particularly in the phase of proteinuria, may benefit from following the Mediterranean diet.

## Figures and Tables

**Table 1 nutrients-13-04110-t001:** Biochemical data during proteinuria and remission.

	Proteinuria	Remission
	mean ± s.d.	mean ± s.d.
uPr/uCr * (mg/mg)	2.81 ± 2.13 *	0.24 ± 0.29
Triglycerides (mg/dl)	117.11 ± 61.72	90.09 ± 50.51
Total cholesterol (mg/dl) *	204.77 ± 52.75 *	158.16 ± 41.16
HDL cholesterol (mg/dl)	69.83 ± 23.26	62.10 ± 16.80
Total protein (g/dl) *	5.66 ± 0.71 *	6.72 ± 0.45
Total albumin (g/dl) *	3.33 ± 0.82 *	4.51 ± 0.31

uCr: urinary creatinine; uPr: urinary proteins. * *p*-value < 0.05 (*T*-test).

**Table 2 nutrients-13-04110-t002:** Number of patients and responses (Yes/No) to the Kidmed questionnaire.

	Yes	No	Score
(1) Takes a fruit or fruit juice every day?	30	18	Yes + 1/no 0
(2) Has a second fruit every day?	20	28	Yes + 1/no 0
(3) Has fresh or cooked vegetables regularly once a day?	34	14	Yes + 1/no 0
(4) Has fresh or cooked vegetables more than once a day?	21	27	Yes + 1/no 0
(5) Consumes fish regularly (at least 2–3 times per week)?	21	27	Yes + 1/no 0
(6) Goes more than once a week to a fast-food (hamburger) restaurant?	5	43	Yes − 1/no 0
(7) Likes pulses and eats them more than once a week?	13	35	Yes + 1/no 0
(8) Consumes pasta or rice almost every day (5 or more times per week)?	38	10	Yes + 1/no 0
(9) Has cereals or grains (bread, etc.) for breakfast?	31	17	Yes + 1/no 0
(10) Consumes nuts regularly (at least 2–3 times per week)?	12	36	Yes + 1/no 0
(11) Uses olive oil at home?	44	4	Yes + 1/no 0
(12) Skips breakfast?	12	36	Yes − 1/no 0
(13) Has a dairy product for breakfast (yoghurt, milk, etc.)?	35	13	Yes + 1/no 0
(14) Has commercially baked goods or pastries for breakfast?	29	19	Yes − 1/no 0
(15) Takes two yoghurts and/or some cheese (40 g) daily?	6	42	Yes + 1/no 0
(16) Takes sweets and candy several times every day?	9	39	Yes − 1/no 0

**Table 3 nutrients-13-04110-t003:** Kidmed score values in the clinical groups of Idiopathic nephrotic syndrome (INS).

	SD	SR	Remission	Proteinuria	SD in Remission	SD with Proteinuria	SR in Remission	SR with Proteinuria
KIDMED score	5.00 ± 2.48	5.37 ± 1.68	5.31 ± 2.36	4.56 ± 2.35	5.40 ± 2.40	3.60 ± 3.60	5.50 ± 1.80	5.25 ± 1.58

**Table 4 nutrients-13-04110-t004:** Fatty acid profile of patients with low or high adherence to the Mediterranean diet.

	Low Adherence (Kidmed Score 2.56 ± 1.21) *n* = 16	High Adherence (Kidmed Score 7.67 ± 0.72) *n* = 15	
	mean ± s.d.	mean ± s.d.	*p*-value
16.0 (Palmitic acid)	23.01 ± 1.57	22.99 ± 1.90	0.97
16:1 (Palmitoleic acid)	0.79 ± 0.34	0.85 ± 0.37	0.65
18:0 (Stearic acid)	11.34 ± 1.80	12.10 ± 2.58	0.35
18:1n9 (Oleic acid)	17.21 ± 2.73	17.53 ± 2.17	0.72
18:1n7 (Cis-vaccenic acid)	1.21 ± 0.22	1.29 ± 0.29	0.35
18:2n6 (Linoleic acid)	25.15 ± 5.34*	21.54 ± 4.46	0.05
18:3n3 (Linolenic acid)	0.22 ± 0.15	0.20 ± 0.09	0.67
20:3n9 (Mead acid)	0.13 ± 0.13	0.18 ± 0.28	0.54
20:3n6 (DGLA)	1.65 ± 0.39	1.53 ± 0.28	0.32
20:4n6 (Arachidonic acid)	11.33 ± 2.01	11.63 ± 1.80	0.66
20:5n3 (EPA)	0.19 ± 0.17	0.21 ± 0.16	0.75
22:0 (Behenic acid)	0.80 ± 0.53*	1.29 ± 0.51	0.01
22:5n3 (DPA)	0.66 ± 0.23	0.77 ± 0.17	0.12
24:0 (Lignoceric acid)	1.80 ± 0.67*	2.39 ± 0.75	0.02
22:6n3 (DHA)	2.39 ± 0.77	2.84 ± 0.65	0.08
24:1 (nervonic acid)	2.01 ± 0.50	2.47 ± 0.87	0.07
SFA	36.96 ± 2.84	38.77 ± 4.74	0.20
MUFA	21.22 ± 2.98	22.15 ± 1.95	0.31
PUFA	41.72 ± 3.80 *	38.90 ± 3.55	0.04
N3	3.45 ± 0.91	4.03 ± 0.74	0.06
N6 *	38.14 ± 4.18	34.70 ± 3.77	0.02

DGLA = Dihomo-gamma linoleic acid; EPA = Eicosapentaenoic acid; DPA = Docosapentenoic acid; DHA = Docosahexaenoic acid; SFA = saturated fatty acids; MUFA = monounsaturated fatty acids; PUFA = polyunsaturated fatty acids; N3 = total omega-3 fatty acids; N6 = total omega-6 fatty acids.* *p*-value < 0.05 at *T*-test.

## Data Availability

The data presented in this study are available on request from the corresponding author. The data are not publicly available due to involve minors.

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
