# Peer review of "Adherence to the Mediterranean Diet Improves Fatty Acids Profile in Pediatric Patients with Idiopathic Nephrotic Syndrome"

_nutrients, 2021, doi:10.3390/nu13114110_

Round 1

Reviewer 1 Report

The authors reported a few papers on of lipid profile of nephrotic syndrome (Refs 2 & 14). The submitted manuscript focused on the relationship between serum lipid profile and Mediterranean diet in children with proteinuria or remission of nephrotic syndrome. The number of patients is different by paper and seems to be overlapped.

What do children eat in low adherence? The Kidmed questionnaire is simple and could not give information on dietary intake. Duration of illness or other anthropometric data like body weight or fat volume would provide additional information.

The question remains whether the lipid profile is a reflection of diet or condition of disease. The Low n6/n3 ratio may be a reflection of a Mediterranean diet. The significant difference was present in only a few fatty acids, but some combinations would appear more clearly if they employed a chief component analysis.

It is better to add all chemical names in Table 4 to fit words in the text.

The number of references in the reference is doubled.

Author Response

The Authors are grateful to the Reviewers for their comments and suggestions, which have contributed to significantly improving the article.

Responses to Reviewer 1

>>>The submitted manuscript focused on the relationship between serum lipid profile and Mediterranean diet in children with proteinuria or remission of nephrotic syndrome. The number of patients is different by paper and seems to be overlapped.

Thanks for your comment. Patients of this study partially overlap with those of the other study, but this is not the case for the data. We chose to obtain new laboratory data to assure far more precise correlation with the Kidmed score. We avoided to correlate old data regarding fatty acid profile to new data regarding mediterranean diet.

>>What do children eat in low adherence? The Kidmed questionnaire is simple and could not give information on dietary intake.

According to the Kidmed questionnaire structure, low adherence did not depend on what they ate, but on what they didn’t eat. As reported in line 210 of the manuscript, low adherence was mainly due to not having breakfast and eating industrial products.

>>Duration of illness or other anthropometric data like body weight or fat volume would provide additional information.

Thanks for the suggestion. Body weight data are now added to the text. The best way to assess fat mass is densitometry, but this implies radiation and could not be prescribed in our cohort.

>>The question remains whether the lipid profile is a reflection of diet or condition of disease.

Nice question. Probably both, but it is still unknown the relative weight of one factor compared to the other. Our present research is focused on understanding the exact role of renal disease and diet on lipid profile.

>>The Low n6/n3 ratio may be a reflection of a Mediterranean diet. The significant difference was present in only a few fatty acids, but some combinations would appear more clearly if they employed a chief component analysis.

Thanks for the suggesting this kind of statistical analysis. The resulting data are now added to the text.

>>It is better to add all chemical names in Table 4 to fit words in the text.

Thanks for the comment. Chemical names have been added to the table.

>> The number of references in the references is doubled.

We have checked the References and they do not appear doubled in our file.

Reviewer 2 Report

The clinical study by Turolo, et al. is a cross-sectional analysis of pediatric patients with idiopathic nephrotic syndrome (INS). The authors have co-related the effect of the Mediterranean diet on the levels of fatty acids and highlight the beneficial shift in the balance of omega 3 and 6 fatty acids. Authors conclude that the Mediterranean diet increases omega-3s and reduces blood linoleic levels, which are considered anti-inflammatory and may benefit INS patients. This study is informative and may interest physicians and dieticians in nephrology. There are some concerns with the study that are enlisted below.

It is unclear whether the increase in omega3 levels is negatively co-related with INS pathology/progression and whether this observation has any clinical/biological importance. Do the patients having relapse have lower omega3 even if they adhere to the Mediterranean diet? And the patients in remission have higher levels of omega3 regardless of whether or not they adhere to the Mediterranean diet?

Please discuss the inclusion and exclusion criteria in the methods section.

Authors have not analyzed the differences in male vs female and stated that “No analysis by sex and gender was conducted, as previous studies have not found sex and gender differences in the adherence to the Mediterranean diet,” and cited reference number 8. I was not able to find any such detail in the respective reference. Please cite references related to the Mediterranean diet responses in males vs females to justify this point. Also, this may be the case in a normal pediatric population. But is it the same case in INS patients? I would recommend analyzing male vs female differences. 

Author Response

The Authors are grateful to the Reviewers for their comments and suggestions, which have contributed to significantly improving the article.

Responses to Reviewer 2

>>It is unclear whether the increase in omega3 levels is negatively co-related with INS pathology/progression.

Thanks for the opportunity to clarify this point. Statistical analysis and also our previous data (Stefano Turolo, Alberto Edefonti, William Morello, Carlo Agostoni, Valentina Manco, Denise Pires Marafon, Maria Rosa Grassi, Luciana Ghio, Marie Louise Syren, Giovanni Montini, FP813
N3 AND N6 PUFA CORRELATE WITH PROTEINURIA IN CHILDREN WITH IDIOPATHIC NEPHROTIC SYNDROME, Nephrology Dialysis Transplantation, Volume 34, Issue Supplement_1, June 2019, gfz106.FP813 ) indicate that omega-3 levels are negatively correlated with proteinuria. This is better specified in line 225 and a “minus” sign was added to the R2 in the text, at line 192

>>and whether this observation has any clinical/biological importance.

Literature data indicates that INS is characterized by the presence of an altered immunological/inflammatory state and it is well known that omega-3 have anti inflammatory properties. As a consequence, a low level of these PUFAs could negatively contribute to INS progression and delay the remission of proteinuria. This sentence has been added to the Discussion.

Do the patients having relapse have lower omega3 even if they adhere to the Mediterranean diet?

Nice question. Unfortunately, we don’t have enough patients to perform a statistical analysis regarding this point. Anyway, as reported in the text, at lines 253 what is really important is the degree of proteinuria, more than the state of “remission” vs “relapse”.

>>And the patients in remission have higher levels of omega3 regardless of whether or not they adhere to the Mediterranean diet?

Even if our cohort was not so large, we were able to perform in this case further statistical analysis on 8 and 11 patients and this new data confirm that a low adherence to Mediterranean diet is associated with lower levels of DHA and total omega3, in contrast to the patients with high adherence. Data are now added in the Discussion section.

>>Please discuss the inclusion and exclusion criteria in the methods section.

Thanks for the suggestion. Inclusion and exclusion criteria have been added to the text.

>>Authors have not analyzed the differences in male vs female and stated that “No analysis by sex and gender was conducted, as previous studies have not found sex and gender differences in the adherence to the Mediterranean diet,” and cited reference number 8. I was not able to find any such detail in the respective reference. Please cite references related to the Mediterranean diet responses in males vs females to justify this point. Also, this may be the case in a normal pediatric population. But is it the same case in INS patients? I would recommend analyzing male vs female differences.

Thanks for the opportunity to clarify this point. By mistake we reported a wrong reference, now the list is corrected with reference 7. However, according to the Reviewer’s suggestion, we performed a statistical analysis on sex and gender differences. No statistical difference was found and the result is now included in the text.